# Rebuilding Gaza's health system: A qualitative study of healthcare workers' experiences and lessons learned from responding to mass casualty incidents (2018–2021)

Anas Ismail[1,2]*, Moatasem Salah[3], Mads Gilbert[4], Yousef H. Abu Alreesh[3], Craig Jones[2]

**1** Population Health Sciences Institute, Faculty of Medical Sciences, Newcastle University, Newcastle Upon Tyne, United Kingdom, **2** School of Geography, Politics and Sociology, Newcastle University, Newcastle Upon Tyne, United Kingdom, **3** Ministry of Health, Gaza, Palestine, **4** Clinic of Emergency Medicine, University Hospital of North Norway, Tromso, Norway

* a.m.s.ismail2@newcastle.ac.uk

## Abstract

### Background

Gaza has faced numerous military attacks that resulted in mass casualty incidents (MCIs). The ongoing genocide in Gaza has destroyed much of the health system, including killing and injuring of hundreds of health care workers (HCWs). Current thinking on the health system reconstruction lacks empirical data and local HCWs' perspectives. The study analyses locally driven innovations and lessons learned by HCWs who responded to MCIs between 2018 and 2021 to guide current and future planning of the reconstruction of the health system in Gaza.

### Methods

This was a qualitative study using online and face-to-face interviews with HCWs who responded to the Great March of Return and the 2021 Israeli military attacks. Transcripts and extensive notes from the interviews were recorded and analyzed on NVivo using thematic content analysis. We used the health system building blocks as themes for deductive analysis with a seventh place-based theme (Gaza-specific) to account for the context of Gaza and the MCIs.

### Results

Problems faced by HCWs mostly related to the nature and complexity of traumatic injuries, shortages in HCWs, particularly specialist doctors, poor coordination among actors, duplication of services, and shortages of supplies and equipment. Locally driven innovations and solutions included establishing new services centers, opening and expanding training programs, starting new coordination bodies, and task shifting

**Data availability statement:** The study is based on qualitative data, which we are not able to make fully available without restrictions. Data access requests should be sent to Dr Craig Jones: craig.jones@newcastle.ac.uk. People and organizations can be easily identified from the transcripts and notes of the interviews. Therefore, making the data fully available without restrictions may divulge the identity of individuals or the organizations in which they worked. Such breach of confidentiality is against the health research ethics approval obtained from the Helsinki committee of the Palestinian Health Research Council. Moreover, due to the current situation in Gaza and the unprecedented attacks on healthcare workers and on organizations working in the health sector, we cannot risk putting any individual or organization, who may be identified through the transcripts, at risk. Hence we cannot make the data fully available.

**Funding:** This study was funded through a fellowship awarded by UK Research and Innovation (UKRI) to Dr Craig Jones as part of the project 'The afterlives of war-related injury: Mapping civilian trauma pathways in conflict settings' (UKRI Future Leaders Fellowship grant number MR/X035794/1). The funders had no role in study design, data collection and analysis, decision to publish, or preparation of the manuscript.

**Competing interests:** The authors declare no competing interests.

of staff and facilities. Lessons learned included strengthening training and employment opportunities for staff, enhancing emergency preparedness and capacities, maintaining coordination bodies, enhancing community engagement and strengthening the governance of the Ministry of Health.

## Conclusion

Reconstruction of Gaza's health system needs to be grounded in its political context and in the experiences of HCWs who have worked in and managed the system. Locally driven solutions and lessons learned can ensure that reconstruction serves as a vehicle for self-determination and sovereignty, rather than entrenching dependency.

---

## Introduction

The attacks on the Gaza Strip since October 2023 have affected all aspects of life for the 2.2 million Palestinians living there. On 2 October 2025, the Palestinian Ministry of Health (MOH) reported 66,225 people killed and 168,938 injured [1]. The actual death toll is likely much higher, with one study suggesting up to 41% more people have been killed [2,3]. Health facilities and healthcare workers (HCWs) have been systematically targeted, resulting in the destruction of health infrastructures, the closure of hospitals, and loss of HCWs [4]. These dire conditions have been exacerbated by a long-lasting blockade, now a complete siege, which has caused systemic economic hardship and has significantly curtailed the availability of health services, causing an unknown number of indirect deaths [5].

The damage caused to the health system will take years and likely decades to repair and reconstruct. This will further limit the availability of health services long after the fighting stops. A recent assessment by the United Nations, the European Union and the World Bank estimates that around $US7 billion is required for the reconstruction of the health sector, with $US 4.3 billion needed in the next three years (2025–2027) [6].

Reconstruction in post-conflict settings faces many challenges. In Bosnia and Herzegovina, for example, reconstruction efforts were still underway more than a decade after the end of the war [7]. A study of health system reconstruction in Cambodia, Afghanistan and Mozambique found similar problems across the three countries: weak governance, a proliferation of uncoordinated actors (particularly non-governmental), widespread destruction of health infrastructure, and reconstruction efforts hampered by continuing political instability [8]. Rwanda provides a good counter example where the determination of national leaders and health system champions helped navigate (post)genocide tensions to build a health system that is among the best in Africa [9,10]. There are also problems related to international donor pressure and influence and the terms of reconstruction, which can echo colonial governance structures such as US President Donald Trump's much-touted

"Gaza Riviera" [11]. Health system reconstruction can thus entrench dependency rather than serve as a vehicle for self-determination and sovereignty.

The MOH has responded to numerous major Israeli military attacks that repeatedly damaged health facilities and infrastructure over the last 20 years [12–14]. During these attacks, HCWs have also responded to many mass casualty incidents (MCIs). This meant that the MOH has led continuous efforts, first, to manage MCIs and, second, to coordinate and implement reconstruction efforts [15,16]. Yet, the ongoing war in Gaza is unprecedented in terms of the brutality and scale of human loss and intentional damage to buildings and infrastructure, with many legal scholars and organizations calling it a genocide [17–20].

Planning for the reconstruction of the health system in Gaza has already begun, with scholars and practitioners proposing diverse ways of 'rebuilding', 'reconstructing', and 'reviving' the health system after the fighting stops [21–26]. Except for the Gaza Health Initiative [24,27], these proposals have taken the form of editorials and commentaries, offering high-level vision and personal views on the principles that should be followed in any reconstruction effort. Missing from these proposals is empirical evidence that draws on the experience of Palestinian HCWs who have worked and managed the health system for many years. Our study aims to fill this gap by providing evidence and insights from those with direct experience of managing the health system during and after repeated MCIs and other health emergencies in Gaza.

We conducted this study in 2022 following two major MCIs in Gaza: The Great March of Return (GMR: 30 March 2018 to 27 December 2019) and Israeli military attacks from 10–21 May 2021. The GMR was a series of community protests by Palestinians in Gaza against the military blockade. The protests took place on a weekly basis (on Fridays), and sometimes twice a week, and were centred around five locations in close proximity to the Gaza-Israel barrier. The GMR resulted in 36,314 casualties (214 killed, 36,100 injured) [28]. The 2021 Israeli military attacks on Gaza resulted in 2,019 Palestinian casualties (259 killed, 1,760 injured) [29].

In this study, our aim is to provide guiding recommendations for current and future planning around the reconstruction of the health system in Gaza based on the experiences of HCWs who have responded to MCIs. These recommendations are drawn from system-level insights shared by HCWs who responded to the GMR and 2021 military attack and are grounded in the political realities of Gaza.

## Methods

### Study design and participant recruitment

We conducted a qualitative study using online and face-to-face semi-structured interviews. An interview guide was developed to explore the problems faced by local HCWs, the solutions and innovations they implemented, and lessons learned when responding to the GMR and the 2021 military attack.

Participants were recruited using purposive and snowballing sampling. The recruitment of participants as well as conducting the interviews took place between 15/07/2022 and 20/09/2022. We aimed to recruit Palestinian HCWs with a wide range of expertise and areas of responsibility from pre-hospital to hospital and finally rehabilitation settings. HCWs who were directly involved in responding to one or both MCIs were invited for interview. An initial list of invitees was identified through the professional networks of the lead co-authors (AI and CJ) as well as reviewing relevant peer-review and grey literature, including meeting minutes of the working groups and taskforces under the Health Cluster – the group of organizations that work in the health sector headed by the World Health Organization (WHO) and MOH. An invitation to participate was first sent, providing a summary of the research, its objectives and details about participation. Invitees who indicated willingness to participate in the study were sent a detailed participant information sheet and a time for the interview was agreed with them. Each participant signed a written a informed consent form after they were given the chance to ask questions about the study and their participation. After each interview, participants were asked to nominate other HCWs to be invited for an interview.

During the interview process it became apparent that our research efforts would benefit from the expertise of one of the recruited participants, and MS was approached. He agreed to join the research team in the analysis phase and is co-author. Although the research was not initially designed to be participatory, the inclusion of MS in the project team would significantly enhance the usefulness of the findings to the MOH, which was a key research priority.

The final sample size was determined by a combination of data saturation and the availability of participants' time given their demanding work schedules. The final list of participants included more frontline HCWs (doctors, nurses, paramedics, physiotherapists), who are naturally in larger proportion compared to HCWs in managerial and leadership positions and in NGOs, meaning the sample was somewhat representative of HCWs involved in MCI response. Also, the list included multiple HCWs from different surgical specialties which were the main specialties involved with the response to MCIs. In turn, and given that surgical specialties are often male-dominated, the list of participants we invited and those who participated were predominantly male. Ethical approval was obtained from the Helsinki Committee of the Palestinian Health Research Council (PHRC/HC/1152/22).

## Procedures

Interviews were conducted either online or in-person, based on the preference of the interviewees. Interviews lasted 40–120 minutes and explored three topics: 1) challenges faced by Palestinian HCWs in the Gaza Strip during and after two MCIs between 2018 and 2021, 2) solutions, innovations and changes to trauma care pathways implemented during and after the MCIs, and 3) systemic lessons learned.

Online interviews were recorded, transcribed, and stored on a password-protected cloud server accessible only to the lead researchers (AI and CJ) of the study. Extensive notes and direct quotes were taken during in-person interviews. Online interviews were conducted either in Arabic or English according to the interviewees' preferences, while all in-person interviews were conducted in Arabic.

## Data analysis

The initial coding of the interviews revealed that participants were discussing challenges, innovations and lessons learned that had health system-wide resonance beyond planning and responding to MCIs. Given the different roles and levels of responsibility the participants had, they spoke about different parts of the health system in which they were involved. Capturing their experiences required using a framework that had a health systems lens. We therefore used the WHO health system building blocks [30] – service delivery, human resources for health, leadership and governance, medical supplies, equipment and essential medicines, health information systems, and health financing – as guiding themes in the deductive content analysis through which we analysed the interviews. This enabled a useful categorization of the various insights shared by participants of all levels of responsibility, from frontline workers in the pre-hospital setting to senior officials in leadership positions.

While there are six conventional building blocks, the data alerted us to a seventh place-based theme that we call 'Gaza specific'. Inspired both by the place-specific data that emerged from the interviews and by literature in health geography that has shown important connections between health and place [31], the Gaza specific category emerged as a way of capturing that which could not easily be contained in the conventional six building blocks. While each state or region has its own specific health-systems challenges, the challenges in Gaza are particularly acute and therefore worth additional scrutiny. Such challenges include prolonged periods of active conflict and Gaza being under siege since 2007, together with restrictions on mobility for both patients and healthcare professionals. Adding a seventh 'Gaza specific' building block became especially useful in allowing us to analyse data that was specific to Gaza, and was also especially helpful in devising recommendations for reconstructing the health system in Gaza.

Interviews were coded using NVivo (©Lumivero). The content of the interviews was first coded independently by two co-authors (AI and CJ), and the coding was compared and any differences reconciled. The final list of codes and their categorization under the building blocks was then discussed with the other authors.

The funders had no role in study design, data collection and analysis, decision to publish, or preparation of the manuscript

## Findings

### Participants characteristics

We invited 32 HCWs (3 females, 29 males), all Palestinian but one, to participate in the study. Nineteen (response rate 59.4%) agreed to be interviewed, and 17 were males (89.5%). Those who did not participate either did not respond to our invitation or declined citing their busy schedule. Table 1 shows the characteristics of the 19 participants. Seventeen participants (89.5%) responded to both the GMR and 2021 attacks. Twelve worked for MOH (63.1%), five (26.3%) worked for international or local non-governmental organisations (NGOs), while one participant worked at the Palestinian Red Crescent Society and one at the Military Medical Services. Six (31.6%) participants held managerial or project officer roles, and 13 participants (68.4%) had clinical roles working directly with patients. One foreign participant was included due to the central role of his organization in the response to both the GMR- and 2021-attacks.

### Challenges

Fig 1 shows the problems reported by HCWs under each of the seven themes. We discuss the most frequently reported one or two challenges under each theme.

   **Service delivery.** Service delivery challenges emanated from the extremely high number of acute injuries caused by various Israeli weapons used, the nature and geography of the MCIs, the effects of MCIs on the delivery of routine

**Table 1. Characteristics of the 19 interviewees.**

| | Gender | Area of work | Place for Work | MCIs responded to | Type of interview |
|---|---|---|---|---|---|
| 1 | Male | Physiotherapy | MOH | GMR and 2021 | Online in English |
| 2 | Male | Emergency Medical Services | PRCS | GMR and 2021 | Online in Arabic |
| 3 | Male | Pre-hospital trauma care | MOH TSP | GMR | Online in Arabic |
| 4 | Male | Management/leadership | Local NGO | GMR and 2021 | Online in English |
| 5 | Male | Project Officer | International NGO | GMR and 2021 | Online in English |
| 6 | Male | Plastic surgery | International NGO | GMR and 2021 | Online in English |
| 7 | Male | Management/leadership | MMS | GMR and 2021 | Online in Arabic |
| 8 | Male | Hand surgery | MOH | GMR | In-person in Arabic |
| 9 | Female | Management/leadership | International NGO | GMR and 2021 | Online in English |
| 10 | Female | General surgery | MOH | GMR and 2021 | Online in Arabic |
| 11 | Male | Surgical nurse | MOH | GMR and 2021 | Phone in Arabic |
| 12 | Male | Emergency medicine | MOH | GMR and 2021 | Online in Arabic |
| 13 | Male | Orthopaedic surgery | MOH | GMR and 2021 | Online in Arabic |
| 14 | Male | Emergency response and coordination | International NGO | GMR and 2021 | In-person in English |
| 15 | Male | Management/leadership | MOH | GMR and 2021 | In-person in Arabic |
| 16 | Male | Orthopaedic surgery/ limb reconstruction | MOH | GMR and 2021 | Online in English |
| 17 | Male | Orthopaedic surgery | MOH | GMR and 2021 | In-person in Arabic |
| 18 | Male | Vascular surgery | MOH | GMR and 2021 | In-person in Arabic |
| 19 | Male | Intensive care unit | MOH | GMR and 2021 | In-person in Arabic |

EMS: Emergency Medical Services, GMR: Great March of Return, MMS: Military Medical Services, MOH: Ministry of Health, NGO: Non-governmental organizations, OR: Operation Room, PRCS: Palestinian Red Crescent Society, TSP: Trauma Stabilization Point.

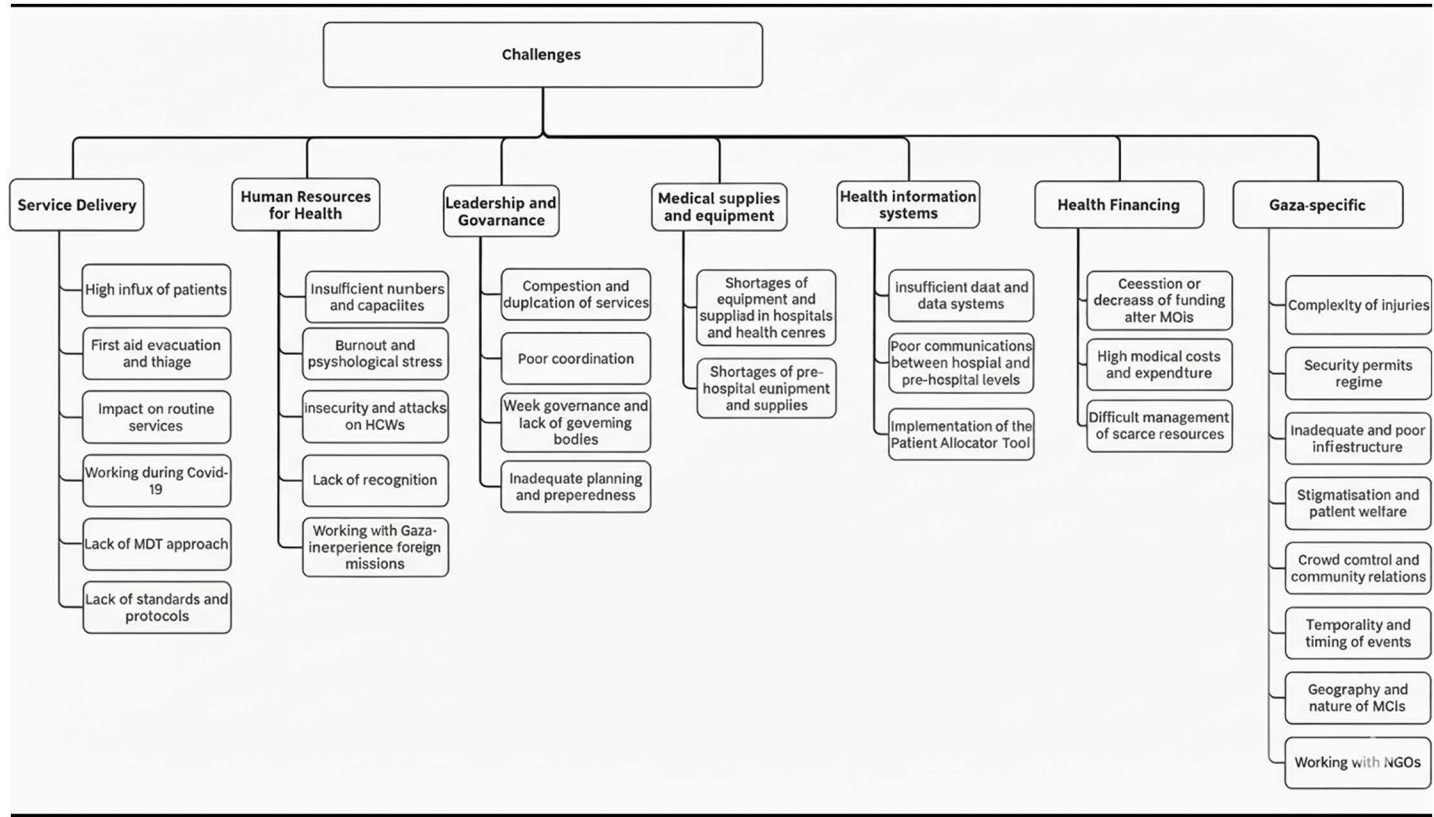

**Fig 1. Challenges reported by the participating HCWs categorized by the seven themes (six building blocks and Gaza-specific).**

services, working during the Covid-19 pandemic, the lack of a consistent multi-disciplinary approach, and the absence of standard protocols and guidelines.

The most frequently reported problem was the high number of acute trauma patients who needed specialized clinical services within a short period of time. While a high influx of patients is characteristic of an MCI, the number of injuries, for example, from the GMR was beyond all expectation and overwhelmed the capacity of the health system in Gaza. Moreover, this was not a one-off incident, but a cumulative series of weekly events as explained by a plastic surgeon:

> *"Each Friday… you find yourself with 200 new cases for Shifa Hospital, for example. There are other cases go to Middle Area and to the North and South. […] maybe 50 cases need admission, and the others can go home […] so these 50 patients have severe injury and they need multiple surgeries and cannot be cured in one week. And the next Friday you find yourself with another 200 cases, and maybe 50 patients need admission."* (Participant 6)

The response to the emergency needs during the GMR and the 2021 attacks forced the termination or a significant decrease in routine services, such as elective surgeries and outpatient clinics. Facilities and human resources offering routine services were often shifted to care for the acutely injured cohort of patients from the GMR and 2021 attacks. One NGO worker explained: *"…[a] huge number of patients [had] operations postponed and they waited [a] very long time for their treatment when the surgeries were resumed."* (Participant 5)

**Human resources for health (HRH).** Problems related to HRH ranged from shortage in staff, particularly specialized doctors, attacks on HCWs, burnout and feeling undervalued, and difficulties working with foreign missions. Shortage of specialist doctors, especially vascular and limb reconstruction surgeons, was the most prominent constraint as they were required to deal with the complex injuries discussed above. The shortages, in turn, led to burnout and stress among the HCWs, some of whom had worked under difficult conditions for over 15 years on only a partial salary. Again, the weekly nature of the GMR, which was followed by the 2021 attacks, meant HCWs could not take a break:

*"[W]e have exhausted the staff physically, psychologically, mentally and socially. That puts great pressure on the staff themselves. […] And you couldn't give them a vacation." (Participant 1)*

Attacks on HCWs was also a major threat. During the GMR, this threat was more pronounced as HCWs were often attacked, injured, and even killed while providing first aid and transporting patients. A paramedic spoke of his experience:

*"..we always tried to exercise caution by standing away from protesters, wearing our marked clothes, and clearly showing our emblems to show that we're only there to provide humanitarian services." (Participant 2)*

## Leadership and governance

Poor coordination, difficulties in preparedness and planning, weak governance and competition and duplication among service providers were the challenges faced in this building block. All four challenges are related, and their causes can be attributed to the chronic state of emergency, as one MOH official explained:

*"You have the start of the blockade and then the 2008-2009 attacks and then 2012, 2014, 2021, COVID-19, 2022. All we have been doing is responding to one emergency after another after another... There is no way to develop a healthcare system in these conditions." (Participant 15)*

The MCIs drew diverse actors, both governmental and non-governmental, but the number of actors made it particularly difficult to coordinate, particularly in light of weak governance structures in Gaza. An NGO worker explained:

*"There were different organizations doing basically the same things and benefiting the same people. So instead of utilizing the resources [in] the best way, there were resources that were wasted due to this duplication" (Participant 4)*

## Medical supplies, equipment and essential medicines

Shortage of supplies and equipment was the main limiting issue reported in this building block, with participants reporting a lack of supplies in both hospital and pre-hospital settings. The HCWs, mainly in surgical specialties, had to provide the best possible care in the face of complete lack or severe shortages of equipment and tools. One participant explained the acute-on-chronic situation:

*"This [shortage] is the normal situation for us, even in the normal times. We don't report what's missing or what we need in order to be fulfilled. On the contrary, we rely on what's donated to us. We work with whatever is donated to us." (Participant 10)*

Paramedics, civil defence workers, and others working in the prehospital settings were similarly having problems with equipment shortages. This was particularly pronounced during the 2021 attacks when people were trapped under the

rubble as paramedics and civil defence workers lacked proper search and rescue equipment and had to dig the victims out with their hands and basic tools.

### Health information systems (HIS)

The problems pertaining to HIS were mainly due to insufficiency of data and data management systems, problems in communication channels between hospitals and pre-hospital staff and specific challenges with the Patient Allocator Tool, a database designed to enhance coordination during the GMR. Data insufficiency was particularly an issue for NGOs as one participant explained:

*"… [The] poor health information system [is] a general challenge because when we design a project, we would really want to write down a justification. You would really want to have the accurate number, the accurate figures. Unfortunately, you didn't get those." (Participant 5)*

An HIS problem particular to the GMR was the implementation of a patient database called the *Patient Allocator Tool*. This database was supposed to coordinate the data available to all service providers to minimize duplication and ensure effective coordination. However, the implementation of the tool was not possible due to shortages in human resources, poor governance, and technical difficulties.

### Health financing

Health financing was the building block least discussed by our participants, perhaps owing to the clinical roles of the majority. Three subthemes were identified in health financing: cessation or decrease of funds after MCIs, allocation and management of scarce resources, and high costs of treatment. These issues fed into each other; resources dwindled at a time of elevated needs both in terms of patient numbers and complexity of needs which in turn were costly to treat. Short-term funding for the immediate emergency response proved unsustainable, which is deleterious to both health systems and patient care as explained by an NGO worker:

*"… usually you have increased number of trauma patients and usually the humanitarian actors receive fund[s] as an emergency response for a short time… then they leave the beneficiaries who [are] still in need for long term rehabilitation… the emergency response to projects it causes, for me, a harm for the beneficiary, some of the project because you [leave] the beneficiary without achieving the outcome of the intervention…" (Participant 9)*

### Gaza-specific context (the 'seventh' building block)

All respondents identified problems that are specific to the context of Gaza or the particular MCIs they participated in. The challenge reported most often in this theme, and throughout the study, was the peculiar complexity of traumatic injuries caused by the various Israeli weapons used during the GMR, as was explained by an orthopaedic surgeon: *"During the Great [March of Return] we receive patients who are completely out of our study at school of medicine"* (Participant 16). High velocity Israeli sniper bullets and gunshots created thousands of very complicated injuries to the extremities, bone, blood vessels, nerves and the soft tissues altogether. One of the participants told us:

*"I think the snipers who were there were actually vascular surgeons. They knew exactly where to hit and how to cause maximum damage. The cases required very complicated care and a very long care journey for at least two or two and half years'" (Participant 15)*

Another major set of obstacles were caused by the strict 'security' permit regime and patient referral systems enforced on Gaza by Israeli occupation authorities, which is one of the many ways of maintaining the blockade. This affected the

movement of injured patients requiring treatment abroad, medical missions coming to Gaza, and medical supplies and equipment entering Gaza. A technical officer at WHO explained:

> "..we identified a lack of equipment and supplies [so] we started the process of procuring [from abroad] but this is the issue… it took almost a year until some of these things end up in the hospitals" (Participant 14)

### Solutions, innovations, and lessons learned

**Service delivery.** New services were created and existing ones expanded in response to the high numbers of injuries and their complexity. Examples included the expansion of the limb reconstruction unit, originally established in 2014 at Shifa hospital with support from Medical Aid for Palestine (MAP), and transferred to Nasser Hospital in Khan Younis to allow its expansion during the GMR. An osteomyelitis laboratory was established alongside so testing could be performed locally. A second limb reconstruction unit was also created in parallel at a non-governmental hospital, Al Awda, in the North of Gaza in 2018, with support from Doctors Without Borders (MSF) Belgium. The Artificial Limb and Polio Centre (ALPC) also expanded its prosthetic and rehabilitation services and opened new rehabilitation centres:

> "..we decided to go immediately for home adaptation for the ones who have mobility disability because we found most of the patients are youth and if they stay at home they may become very depressed" (Participant 4)

Surgical care was enhanced through cooperation between the MOH, non-governmental hospitals, and NGOs to move some routine services and surgical follow-up care of the wounded to the less busy non-governmental hospitals as one surgeon explained:

> "Typically, when I operated on a patient, I would see him only after two weeks or even three weeks. Imagine if this patient left to home without this role of these NGOs… in severe infection and you have to amputate. But they did the great job in dressing and giving antibiotics and they replaced the role of the governmental hospital at that time actually." (Participant 16)

Other innovations included task shifting of people and facilities to support high-demand services, improving first aid and triage in the field and in the hospitals, improving clinical care in the trauma stabilisation points (TSPs) and hospitals, and initiating multi-disciplinary teams to provide comprehensive care to patients. The internal medicine emergency department was transformed to a surgical emergency department on protest days to accommodate the high number of wounded anticipated. Likewise, the daycare surgery unit at Al Shifa hospital was transformed to a second ICU unit to accommodate the number of patients requiring intensive care. In the prehospital phase, ambulances were adapted to carry several patients to shorten evacuation delays times. This helped improve the clinical care provided to patients over time during the GMR, including working in multi-disciplinary team (MDT) as a plastic surgeon explained:

> "(F)or each patient, he's seen by a multidisciplinary team including the orthopaedic doctor, the plastic surgery doctor, the qualified nurse, the physiotherapy, the mental health so it's a holistic approach" (Participant 6)

The lessons learned overlapped with the innovations with a forward-oriented commitment to improving services for patients during the next period of violent attacks on Gaza. One lesson was to enforce an MDT and in particular not to neglect the delayed psychological aspects of injury. Another lesson our participants stressed was continuing to improve the preparedness for and quality of prehospital care, particularly mass casualty management and the national emergency medical teams (EMT) initiative. In terms of clinical care, it was suggested that other specialised units be established

similar to the one established for limb reconstruction. Finally, there was a strong sense that emergency preparedness needed to be strengthened, including scaling up the infrastructure and capacity for surgical care.

**Human resources for health (HRH).** Capacity building efforts, including starting new specialities and expanding existing ones, was the main innovation implemented in the HRH building block. A vascular surgery specialty training programme was officially initiated during the GMR in response to the acute needs. Furthermore, two trauma-related specialties, emergency medicine and intensive care and anaesthesia, were expanded and new incentives added so more doctors choose these specialties. Additionally, the national EMT initiative was started to train and prepare one or two EMTs to have the qualification and preparedness to respond effectively to MCIs.

While some participants viewed working with foreign missions as a challenge, other participants considered the cooperation with some foreign missions and their support as an important innovation, as illustrated by one of the participants:

*"The surgical missions that were coming to help were very beneficial to us and they were a valuable addition in terms of the work they were doing and the skills they were training us on. They were never a burden." (Participant 13)*

The third innovation implemented in the HRH building block was increasing the number of available staff through employment and incentives for joining the newly established or scaled-up specialty training programmes. The fourth innovation was extending work-shifts so that in-demand services be offered for extended hours. For example, local Palestinian NGO workers temporarily worked two shifts, morning and late afternoon, to meet increased demand.

As for lessons learned, our participants stressed the need to sustain the increase in employment and training opportunities to enhance MOH's staff preparation for any future MCIs and to be able to offer higher quality services overall. Moreover, some participants highlighted the need for these actions to go beyond physicians so that more nurses, health information specialists, and paramedics are employed and trained.

The second lesson learned in HRH was to recognize the efforts HCWs, particularly junior ones, make while working in emergencies and offer compensation to improve their personal and economic situation. One participant illustrated this:

*"it's not really about the money or the financial stuff. it's really about feeling appreciated and feeling that the efforts you have put during the escalation did not go in vain." (Participant 11)*

Finally, a lesson learned is to re-think how different HCWs can be best involved in emergency response planning. These includes involving rehabilitation specialists in the care of patients starting from the emergency room, a senior surgeon being present to lead the triage in the hospital, and tasking foreign missions with patient follow-up and not only work in the operation rooms.

## Leadership and governance

Innovations fell under three subthemes: turning MCIs into opportunities to improve cooperation, creating bodies to enhance governance and leadership, and continuous learning to improve MOH leadership and governance. Improved coordination occurred at different levels and between all types of actors, including different sectors of the government, local actors and NGOs and international NGOs. Despite the difficulties of working during the GMR, it also brought opportunities to bring everyone together as an NGO worker explained: "*[The] GMR really brought everyone together and that was a positive thing that resulted from the GMR, everyone was thinking how to benefit those patients*" (Participant 5).

Improving the coordination occurred through various bodies that were created while responding to the MCIs, particularly the GMR. Examples include the Trauma Working Group and the Rehabilitation Taskforce, which were created under the

Health Cluster umbrella. Moreover, the massive influx of patients and overwhelming pressures on the health system also stimulated the MOH to learn from this experience and increase its emergency preparedness, as explained by an MOH official:

> "I worked through all events, 2008-2009, 2012, 2014, 2021, and the last escalation 2022, I never saw a day like May 14th [2018]. The system was on the verge of collapse and we lost control of everything. The numbers were unimaginable, the number of surgeries was unimaginable, and the patients just kept coming without stop. It [is] a tragedy with the full meaning of the word. This motivated us to concentrate on health system and emergency preparedness and increase our capacities." (Participant 15)

Two subthemes in the lessons learned were identified. First was to maintain the coordination mechanisms and bodies that were created and to bolster these mechanisms. An NGO worker expressed the need for this, *"I would not really give up on the coordination that we have already accomplished with the key players in the field, because this is really a blessing and you know, like, bringing everyone on the same table was really the toughest thing to do in the past"* (Participant 5). Second, participants called for sustained efforts towards systemic emergency preparedness and planning. This includes the MOH assuming stronger governance role, running emergency exercises, and incorporating plans to sustain non-emergency and non-conflict health services to the population.

### Medical supplies, equipment and essential medicines

Innovations in this building block involved improvisation by Palestinian HCWs, particularly when it came to shortage or lack of medical supplies and equipment. First, HCWs, deployed new equipment and supplies to respond to the surge in needs and complex injuries. Examples of such innovation include positioning domestic fridges in the operations department to store blood products for immediate availability during surgery, the use of limb reconstruction equipment that was previously not available in Gaza, and the use of oxygen generating devices in the TSPs to provide oxygen therapy to patients immediately in the field. Second, HCWs did their best to increase the efficiency of available equipment and supplies. One example was restructuring the ambulance compartments to fit up to four patients instead of one, and another example was pharmacists analysing expired medications which were still safe and effective. The innovations implemented in response to MCIs were helpful in future emergencies. For example, the TSPs which were used to respond to the GMR were repurposed and used as respiratory triage centres during Covid-19 pandemic.

Two subthemes of lessons learned were mentioned by participants: make available a wider array of equipment and supplies in hospitals and ensure greater stocks of medical supplies and equipment. HCWs with clinical roles thought more equipment should be supplied to hospitals including point-of-care ultrasound, computed tomography machines and mobile x-ray machines in the emergency department as well as ones for vascular surgery, orthopaedic surgery and intensive care. On the other hand, HCWs with managerial roles reported that they would seek to increase pre-positioned stocks of medical supplies and equipment to be enough for at least six months of emergency.

### Health information systems (HIS)

When it came to HIS, the HCWs used new methods of data collection and management to overcome the main constraints during MCIs explained above. This included videotaping patients' injuries in TSPs as well as by specialists to record any improvements or deteriorations. Another example was registering injuries in the field using programmes where workers could register data offline and then upload them when online, as an NGO worker explained:

> "..we have our own program, but we develop[ed] it to be on tablets and you can make an entry without Internet. After that when you connect to the Internet it goes to the database immediately." (Participant 4)

The second innovation was the creation of the Patient Allocator Tool to track patients and coordinate between service providers. However, as explained in the challenges section above, the implementation of the tool proved problematic. A third innovation was the use of Israeli SIM cards to overcome weak Palestinian mobile signals near the borders. This helped improve communications between pre-hospital staff, ambulances, and hospital staff.

Three lessons learned were expressed by the respondents. First is the development of an information system to map the actors and their services to improve efficiency and coordination of service provision. Second is to develop an emergency code that can be sent to all HCWs once an emergency or MCIs happens to create instant awareness of the situation and help HCWs present to the hospital. Third is enforcing a central data collection system rather than having multiple systems run by the different actors. This was explained by a participant:

*"..they [the government] have the unified gate, the information system. [it is] supposed to collect data from all NGO's working for persons with disabilities, providing health services, social services and protection, etc. But it is not filled by the members because they don't feel that they are obliged or enforced by the Ministry of Social Development to use this system." (Participant 9)*

### Health financing

The main health financing innovation that occurred during the MCIs was funding local, non-governmental hospitals to expand their services so they would receive patients and relive some pressure on the MOH. This resulted in improving the infrastructure of these hospitals and increasing their capacities after the MCIs finished. Two participants said their organizations overcame the challenge of decreasing funds after the MCIs finish and managed to sustain their funds and continue their work.

Two lessons learned were expressed by two different participants. One was to align the funding needs of the organizations with the policies of their donors in order to sustain these funds. The other lesson was to improve funding available for staff salary to improve their professional and personal conditions.

### Gaza-specific context (the 'seventh' building block)

The solutions and innovations relevant to the Gaza-specific context came in four subthemes: new ways of support by international NGOs, locally driven logistics solutions, positive community engagement, and patient advocacy and welfare.

Some international NGOs played a positive role by helping open new services, particularly for limb reconstruction, expand existing facilities and infrastructure, such as expanding the emergency department at Shifa hospital that was supported by ICRC, and funding non-governmental hospitals to expand their surgical wards to relieve pressure on the MOH. On the other hand, locally driven solutions were created, for example, to facilitate the establishment of TSPs near to the sites of protest. Large containers were used to store the components of the TSPs nearby to facilitate set-up each week. These solutions were also supported by positive community engagement, whereby community members either contributed equipment and tools or helped in setting up the TSPs and providing first aid.

Finally, as patients had long therapeutic journeys ahead of them, patient-to-patient advocacy were noticed whereby patients were advised against traveling abroad by fellow patients who found no benefit in traveling or by foreign experts who assured patients they were receiving excellent care in Gaza. Finally, as a participant explained, vocational training and sports were utilized to fill patients' lives with positive activities:

*"We try to do vocational training [and] also, we guide them for sport. We made a team for the wounded [for] football for amputees, basketball… so their schedule and their time is full of effective things". (Participant 4)*

In terms of lessons learned, participants pointed out three key lessons. The first was to strengthen the coordination of emergency response through ensuring safe corridors for HCWs and ambulances to move, recruiting foreign missions with previous experience working in Gaza, and ensuring adequate staff are present in hospitals other than Al Shifa hospital and European Gaza Hospital. Linked to the first, the second lesson was to concentrate support on the government health sector, as explained by a participant: "*… this is the sector that is going to last for the community of Gaza, and we shouldn't be really foreign-focused*" (Participant 5) The third lesson was to work on increasing community awareness of how HCWs function in emergencies to achieve higher level of collaboration with the community.

## Discussion

This study has reported the systemic problems and obstacles faced by HCWs who responded to MCIs between 2018 and 2021. It has also reported locally driven innovations and solutions as well as insightful lessons learned by HCWs, which in turn help in the formulation of guiding recommendations for future health system reconstruction efforts in Gaza. Findings related to the GMR are more frequently reported than the 2021 military attacks, because the GMR can be thought of as over 80 MCIs combined since the health system responded to MCIs caused by Israeli attacks on the weekly protests for nearly two years.

The variety of recruited participants in the study, in terms of their nature of work as well as level of responsibility, ensured that we captured challenges, innovations and lessons learned in a comprehensive way. The input provided by our participants ranged from detailed insights from frontline workers in operation rooms, wards and prehospital settings to insights from HCWs with managerial responsibilities, including in the non-governmental sector. Moreover, some of the participants have more than 10 or 15 years of experience in the health system in Gaza and have provided historical context and insight that was helpful in analysing our findings in a systematic manner.

The challenges our respondents reported are not entirely unique. Evidence from previous military attacks on Gaza points to both system-wide problems, such as insufficient high-level coordination and inadequate numbers of specialists, but also specific problems related to hospital infrastructures and communication between different healthcare facilities [30]. Other problems related to burnout, stress, hard working conditions were expressed by Palestinian HCWs both before and during the Covid-19 pandemic [32,33]. These problems are shared by HCWs in other places while responding to MCIs, especially in conflict settings. For example, in Syria, HCWs faced challenges of insecurity and attacks on them and their families, attacks on hospitals, and severe shortages of staff, supplies and consumables [34]. Similarly in Lebanon, HCWs responding to the 2020 Beirut blast faced very challenging circumstances as they suddenly received a huge influx of patients that overwhelmed the human and material resources available [35].

The challenges our participants reported have been made more profound by the direct and deliberate attacks on healthcare over the last two and half years. A prominent challenge will be to rebuild the destroyed health facilities and infrastructure; by June 2025, there were 735 attacks on healthcare affecting 125 health facilities and damaging 34 hospitals [36]. In addition, the remaining hospitals suffered severe shortages in essential medicines with a study finding only four out of 25 essential medicines met the WHO 80% availability benchmark in public health facilities by the summer of 2025 [37]. As for HCWs, over 1,700 were killed, over 300 are detained and unknown number injured [38]. The number of HCWs who fled Gaza is also unknown. These losses cannot be easily or quickly overcome; the training of qualified HCWs, particularly specialist doctors, will take years. This had left patients without proper care during the genocide, with junior doctors and medical students often having to step up and provide clinical care [39].

As for the future of the health system in Gaza, there is a growing body of literature that addresses its reconstruction. This has been mainly in the form of editorials and commentaries whose authors offer their views on important considerations and principles to be taken into account when 'rebuilding', 'reconstructing' or 'reviving' the health system in Gaza [21–26]. These attempts have mainly been 'built on scarce available data and assumption', as one article explains [24]. Our study fills this gap as it draws on the experiences of local HCWs and addresses challenges that existed prior to

October 2023. One thing authors agreed on is that the reconstruction of the health system in Gaza cannot happen unless the flighting stops. As Dr Tedros Adhanom Ghebreyesus, WHO Director General, and Dr Hanan Balkhy, Regional Director for the Eastern Mediterranean region, had said, *"… a permanent cessation of hostilities is needed to provide a conducive environment for reconstruction and restoration"* [21]*.* Many of our respondents made a similar point as they explained that Gaza has been dealing with one Israeli military attack and health emergency after another for decades. This situation seriously hampers health system research, planning and evidence-based development.

Our participants shared several innovations and lessons learned in the service delivery block that can guide future reconstruction efforts. The most frequently reported was the creation of new services and expanding of existing ones to respond to new trends and quantities in traumatic injuries. The ongoing genocide has produced a burgeoning burden of traumatic injuries that require long-term rehabilitation. For example, WHO estimated that 25% of the injuries from the war will require long-term rehabilitation [40] and reports from the rehabilitation task force under the health cluster estimate that there are 30,000 trauma cases that need long term rehabilitation and that nearly a quarter of the amputation cases and 70% of surgical burn cases are paediatric cases [41]. These are new health conditions that will be added to already existing health conditions in the population, including cardiovascular diseases, cancer and congenital malformations.

Therefore, as the health system is rebuilt, clinical and rehabilitation services need to be started and/or expanded, particularly paediatric ones, to meet the needs of tens of thousands of injured. One model to follow is to establish specialised centres and units, as was done for limb reconstruction during the GMR, which helps standardize care, ensures the availability of the best healthcare workers, and facilitates working with foreign missions coming to Gaza. The presence of such specialised centres can also help with patients who would otherwise require referral abroad in two ways. First, it can reduce the number of such referrals as more advanced care can be made available at these centres. Also, it can help streamline and unify the referral process as local and foreign specialists assess who can be treated locally and who needs to be referred and where. Furthermore, the need to adopt an MDT approach and provide holistic care, including mental health care, is something our participants repeatedly stressed. Given the emerging evidence of extremely high levels of mental health disorders among the population in Gaza [42,43], creating MDTs to deliver care to patients holistically is an imperative of any recovery efforts in the service delivery building block. Using telemedicine and drawing on diaspora doctors can be utilized in the MDTs to ensure scarce specialties are covered in the MDTs. An example of support from diaspora and foreign HCWs during the genocide has already proven effective [44] and can be a basis for future scale up.

Our participants shared several insights pertinent to the HRH building block. The most important is offering paid training opportunities through opening new specialty training programmes and expanding existing ones, a recommendation shared by all pieces discussing Gaza's health system reconstruction. This achieves two goals: it helps in expanding the capacity to deliver services as discussed above, and it offers employability and career opportunities for Palestinian HCWs. Evidence from the literature [21–23,25] shows that the latter is crucial given the grave losses in Palestinian HCWs who were killed, injured, detained or forced to flee. Achieving this goal also requires focusing on the younger generations of HCWs, including students, by offering they high quality education and training and opening up opportunities that attract them to stay rather than flee Gaza.

In terms of governance and leadership, our participants stressed the importance of seizing opportunities to create coordination and governing bodies that arise in emergencies and maintain them after the emergency ends, which is a view also shared by Alkhaldi and Alrubaie [23] who called to make the reconstruction efforts an opportunity to utilize new approaches and models for the reconstruction and not simply restoring the old system. In the GMR and 2021 attacks, this happened through the creation of the trauma working group and the rehabilitation task force. Reconstructing the health system in Gaza will undoubtedly involve many actors and entities, and creating mechanisms to coordinate and guide these actors must start now and be maintained after the fighting stops. This principle is one that others have also called for. Blanchet et al [24] suggested building on existing coordination mechanisms, Smith et al [25] warned of the consequences of the absence of such coordination which would result in creating parallel systems that undermine the

pre-existing health system, and Alkhaldi and Alrubaie [23] advocated for coordinating bodies to be inclusive of all disciplines and stakeholders, particularly civil society.

As for health financing, our participants offered limited insights given the clinical nature of the roles most of them had. The insights they offered are still illuminating as they called for aligning funding with local needs and priorities, particularly financing staff salaries to improve their working and life conditions. Alkhadli and Alrubaie [23] proposed starting a financing facility that is overseen by an advisory board and management by an executive body. Abuelaish and Musani [22] advocated for the scaling up of funding for both immediate and long-term health needs. Strategic financing to fund long-term needs is one way to tackle the challenge of cessation of short-term funding reported by our participants. In line of this, Nasari et al [26] advocated for 'investment from international stakeholders beyond short-term emergency relief efforts'.

Relevant to the governance and health financing building blocks discussed above is the need to invest in the public sector. Experiences of the work of Partners in Health work in Haiti after the 2010 earthquake and Rwanda after the 1994 genocide provide excellent examples of reviving and reconstructing the health systems after crises with deliberate engagement with and strengthening of the governmental public infrastructure and services [45]. These experiences link the health financing building block with the leadership and governance one; it is important to ensure emergency and development funding for health services align with the vision and plans set by the MOH as the main service provider in Gaza. Such sound health financing strategies also follow international declarations such as the 2005 Paris declaration on Aid Effectiveness [46] and the 2008 Accra Agenda for Action [47], which called for working with and through the public sector.

The procurement and availability of medical supplies and equipment will be a cornerstone in any immediate relief or long-term reconstructions efforts. Abuelaish and Musani [22] suggest making use of global procurement of medical equipment and supplies that ensure obtaining needed equipment and supplies at best prices. This echoes the insights shared by our participants who expressed the need to make available a wide array of equipment and supplies as needed by all specialties and facilities, particularly as the lack or shortage of such supplies has been the norm in Gaza during the blockade. Global procurement also makes it possible to ensure larger amounts available in stocks to meet surge needs, which was a lesson learned by our participants.

Having a reliable and robust HIS is vital to any reconstruction effort in the health system. Our participants found it important to have systems in place that map and tracks active actors in the health sector and map the services they provide. Currently, WHO offers similar insights through its HeRAMS platform [48]. Such platform and the information it collects should be owned by the MOH as well to give its decision makers real-time information on who is doing what and where. Yaghmaei et al [49] also call for the harmonization of data collection and national ownership of HIS so the MOH in Gaza can rebuild a resilient HIS. As for clinical data, it is pivotal to improve registries of patients so that they are electronic and reliable, which allows the generation, analysis and dissemination of data to inform decisions and policies made with regards to patient care.

Finally, we must stress that post-war health rebuilding in Gaza is never purely technical. It is deeply political: it means confronting the blockade, the destruction of infrastructure, and the denial of Palestinian sovereignty that decide who has access to care, whose expertise is recognized, and how scarce resources are controlled [16,50,51]. A decolonial approach in Gaza must therefore focus not only on health system reconstruction but also, as Alkhaldi and Alrubaie point out [23], on rebuilding all public social systems, including health, education, sanitation, food security, transportation, and others. This cannot be done without tackling the structural determinants of ill-health produced by decades of occupation and siege. Then reconstruction may disrupt, rather than entrench, the colonial patterns of structural violence that have long undermined the right to health in Gaza.

## Limitations

The study has several limitations. First, there is an imbalance in the gender distribution of both the initial list of people we invited to the interviews and the participants who did participate in the study. This has several reasons. This could be due

to a bias in the professional network of the first author who did most of the purposive recruitment and of the interviewees' networks who helped in snowball sampling other participants. Also, it could be a reflection of the gender imbalance in the distribution of HCWs working in MCIs planning and response in the health system in Gaza, particularly that the response is dominated by surgical specialties which are often male-dominated. Second, the study sample is missing HCWs from non-surgical specialties, e.g., internal medicine, pediatrics, and primary healthcare, whose insights and inputs are essential when it comes to reconstructing the health system in Gaza. This was due to the topic of interest of the study being response to MCIs, which mainly relates to HCWs in surgical specialties. Third, unlike commentaries and editorials who can freely discuss all aspects of reconstruction, we are constrained to discussing what our participants reported. Therefore, the discussion of the health system reconstruction is not encompassing of all issued we would discuss if it were a commentary or an editorial. For example, we recognize that primary healthcare should be the foundation of the health system in Gaza, but this aspect is not mentioned by our participants and therefore doesn't feature in our study.

## Conclusion

Our study draws on the experiences of Palestinian HCWs who responded to two major MCIs between 2018 and 2021. Using the WHO's building blocks as a framework, we reported major problems faced by HCWs, local solutions and innovations and lessons learned by HCWs. We used these experiences to draw empirically grounded insights from local expertise to inform future recovery planning, with a focus on service delivery, HRH and governance and leadership. Reconstruction of Gaza's health system needs to be grounded in its political context and in the experiences of HCWs who have worked in and managed the system. Locally driven solutions and lessons learned can ensure that reconstruction serve as a vehicle for self-determination and sovereignty, rather than entrenching dependency.

## Supporting information

**S1 File. Inclusivity-in-global-research-questionnaire.**
(DOCX)

## Acknowledgments

We would like to thank the healthcare workers who agreed to participate in this study and volunteered their time to do the interviews.

## Author contributions

**Conceptualization:** Anas Ismail, Craig Jones.

**Data curation:** Anas Ismail, Craig Jones.

**Formal analysis:** Anas Ismail, Moatasem Salah, Mads Gilbert, Craig Jones.

**Funding acquisition:** Craig Jones.

**Investigation:** Anas Ismail, Moatasem Salah, Mads Gilbert, Craig Jones.

**Methodology:** Anas Ismail, Craig Jones.

**Project administration:** Craig Jones.

**Resources:** Anas Ismail, Craig Jones.

**Software:** Craig Jones.

**Supervision:** Mads Gilbert, Yousef H Abu Alreesh, Craig Jones.

**Validation:** Anas Ismail, Moatasem Salah, Mads Gilbert, Yousef H Abu Alreesh, Craig Jones.

**Visualization:** Craig Jones.

**Writing – original draft:** Anas Ismail.

**Writing – review & editing:** Anas Ismail, Moatasem Salah, Mads Gilbert, Yousef H Abu Alreesh, Craig Jones.

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
