## [Decision Letter · Decision Letter 0]

11 Dec 2025

Rebuilding Gaza's health system: a qualitative study of healthcare workers' experiences and lessons learned from responding to mass casualty incidents (2018–2021)

Dear Dr. Ismail,

We look forward to receiving your revised manuscript.

Kind regards,

Nikolaos Georgantzis

Academic Editor

PLOS One

3. In your Discussion section, the paragraph starting with "Finally, we must stress that post-war health rebuilding in Gaza ... " (lines 659-667) please provide appropriate contextualisation for this section by adding additional literature (and references).

“This study was funded by Newcastle University through a grant awarded to Dr Craig Jones.”

5. In the online submission form, you indicated that [The study is based on qualitative data, which can be available upon reasonable request. We are not able to make the data fully available without restrictions because people and organizations can be easily identified from the transcripts and notes of the interviews. Therefore, making the data fully available without restrictions will likely make us not able to maintain confidentiality of the participants and other persons they mention in the interviews].

Editorial request:

The last paragraph, starting with "Finally" (line 659) and pointing out that the reconstruction has a political context (the denial of Palestinian sovereignty etc.) feels somewhat editorial. We kindly request you give more context to this part of the discussion. If you wish, feel free to support your statement with additional literature.

Reviewers' comments:

Reviewer's Responses to Questions

**Comments to the Author**

1. Is the manuscript technically sound, and do the data support the conclusions?

Reviewer #1: Yes

Reviewer #2: Yes

2. Has the statistical analysis been performed appropriately and rigorously?

Reviewer #1: Yes

Reviewer #2: No

3. Have the authors made all data underlying the findings in their manuscript fully available?

Reviewer #1: No

Reviewer #2: Yes

4. Is the manuscript presented in an intelligible fashion and written in standard English?

Reviewer #1: Yes

Reviewer #2: Yes

Reviewer #1: This is a very well-researched and insightful article that offers a strong understanding of post-conflict settings while highlighting how the situation in Gaza is distinct—particularly given the ongoing genocide and the deliberate targeting of the health sector. The discussion on decolonization is both timely and essential, and I found the integration of a Gaza-specific theme within the health system building blocks particularly smart and innovative. The paper also does an excellent job illustrating the dependence on equipment and resources from NGOs and international actors, as well as the need to reimagine this relationship to better serve the Palestinian health system.

That said, I believe the limitations section could be expanded further, especially to reflect more deeply on the authors’ strategic insights and how participants’ feedback might inform practical implications. While the study is qualitative, incorporating some quantitative context—such as data on hospitals, healthcare professionals, and medical resource shortages—could help ground the analysis more firmly. Additionally, emphasizing the legitimacy and depth of participants’ perspectives would strengthen the argument, given their firsthand experience with the health system’s evolution over time.

Overall, this was an extremely engaging and enlightening piece that sheds light on Gaza’s future—an aspect that often gets overshadowed by the focus on current atrocities.

Reviewer #2: Dear Authors,

I enjoyed reading your paper titled "Rebuilding Gaza's health system: a qualitative study of healthcare workers' experiences and lessons learned from responding to mass casualty incidents (2018–2021)". Your paper contributes to the literature by providing empirical evidence and insights from those with direct experience of managing the health system during and after repeated MCIs and other health emergencies in Gaza. The paper offers valuable guiding recommendations for current and future planning around the reconstruction of the health system in Gaza based on the experiences of HCWs. However, I have a few comments to help improve the paper.

Major Comments:

1. The manuscript uses WHO’s building blocks plus a seventh “Gaza-specific” theme as the framework for analysis. However, the framework is introduced without specific justification. It is unclear how the Gaza-specific category interacts with the other building blocks. How did the Gaza-specific theme emerge inductively from the data?

2. The sample size needs more discussions. Although participants were recruited using purposive and snowballing sampling, the manuscript does not explain whether this recruitment approach yields a sample that is representative of healthcare workers operating in the specific context of Gaza.

3. As mentioned in the previous point regarding sample representativeness, the authors note the gender imbalance in their sample in the Discussion section. However, they do not provide any indication of the extent of gender imbalance among HCWs in Gaza overall, making it difficult to assess whether the sample reflects the underlying population structure.

Minor Comments:

1. Define “building blocks” the first time it appears in the Data Analysis subsection.

2. The figure’s readability could be improved. The authors may consider using white background blocks with black text to enhance contrast and make the information easier to read.

**Do you want your identity to be public for this peer review?** For information about this choice, including consent withdrawal, please see our Privacy Policy

Reviewer #1: **Yes:** Adèle ZITO

Reviewer #2: No

---

## [Author Response · Author response to Decision Letter 1]

23 Jan 2026

We are very grateful to the editors and reviewers for the time spent reading the article and for constructive feedback. We found the reviews both supportive and generative and have used them to sharpen the contribution. The re-submitted article is revised accordingly with all edits in red text. What follows is a full response to reviewers and editor comments.

The revisions are focused on key points raised by the reviewers.

Comments by the editor:

1- Ensuring the manuscript meets PLOS ONE’s requirements

Changes have been made to align the manuscript with PLOS ONE’s requirements.

2- Including a copy of PLOS’ questionnaire on including in global research

A copy of PLOS’ questionnaire is now included in the submission.

3- Provide contextualization by adding additional literature to discussion

A few relevant articles have been cited to support our point that we make in the last paragraph of the discussion:

“it means confronting the blockade, the destruction of infrastructure, and the denial of Palestinian sovereignty that decide who has access to care, whose expertise is recognized, and how scarce resources are controlled (16,49,50). A decolonial approach in Gaza must therefore focus not only on health system reconstruction but also, as Alkhaldi and Alrubaie point out (23), “

4- Stating role of funder"

This is clarified in the paper: “The funders had no role in study design, data collection and analysis, decision to publish, or preparation of the manuscript”

5- Explaining why data cannot be made available

A clarification is now added in the online submission form.

Comments by the reviewers

1. R1 asks to expand the limitations of the study

An additional limitation has been added to acknowledge the lack of representation of some types of HCWs whose insights would be important:

“Second, the study sample is missing HCWs from non-surgical specialties, e.g. internal medicine, pediatrics, and primary healthcare, whose insights and inputs are valuable when it comes to reconstructing the health in Gaza. This was due to the topic of interest of the study being response to MCIs, which mainly relates to HCWs in surgical specialties.”

2. R1 asks to incorporate some quantitative context to the qualitative findings

Quantitative context has been added in a new paragraph

“The challenges our participants reported have been made more profound by the direct and deliberate attacks on healthcare over the last two and half years. A prominent challenge will be to rebuild the destroyed health facilities and infrastructure; by June 2025, there were 735 attacks on healthcare affecting 125 health facilities and damaging 34 hospitals (35). In addition, the remaining hospitals suffered severe shortages in essential medicines with a study finding only four out of 25 essential medicines met the WHO 80% availability benchmark in public health facilities by the summer of 2025 (36). The loss of skilled HCWs is a loss that cannot be overcome. As for HCWs, over 1,700 were killed, over 300 are detained and unknown number injured (37). The number of HCWs who fled Gaza is also unknown. These losses cannot be easily or quickly overcome; the training of qualified HCWs, particularly specialist doctors, will take years. This had left patients without proper care during the genocide, with junior doctors and medical students often having to step up and provide clinical care (38).”

3. R1 asks to emphasize the legitimacy and depth of participants’ perspectives

A new paragraph has been added in the discussion section:

“The variety of recruited participants in the study, in terms of their nature of work as well as level of responsibility, ensured that we captured challenges, innovations and lessons learned in a comprehensive way. The input provided by our participants ranged from detailed insights from frontline workers in operation rooms, wards and prehospital settings to insights from HCWs with managerial responsibilities, including in the non-governmental sector. Moreover, some of the participants have more than 10 or 15 years of experience in the health system in Gaza and have provided historical context and insight that was helpful in analysing our findings in a systematic manner.”

4. R2 asks for an explanation of how the WHO building blocks framework came to be used. They also ask us to explain the “Gaza-specific” building block and reflect on how it emerged and how it interacts with the other building blocks.

We have now added an explanation of why the WHO’s building blocks framework was used and have also accounted for how the Gaza-specific theme emerged. The following paragraphs have been added to the ‘data analysis’ heading of the Methods section:

“Given the different roles and levels of responsibility the participants had, they spoke about different parts of the health system in which they were involved. Capturing their experiences required using a framework that had a health systems lens. We therefore used the health system building blocks (30) – service delivery, human resources for health, leadership and governance, medical supplies, equipment and essential medicines, health information systems, and health financing – as guiding themes in the deductive content analysis through which we analysed the interviews. This enabled a useful categorization of the various insights shared by participants of all levels of responsibility, from frontline workers in the pre-hospital setting to senior officials in leadership positions.

While there are six conventional building blocks, the data alerted us to a seventh place-based theme that we call ‘Gaza specific’. Inspired both by the place-specific data that emerged from the interviews and by literature in health geography that has shown important connections between health and place, the Gaza specific category emerged as a way of capturing that which could not easily be contained in the conventional six building blocks. While each state or region has its own specific health-systems challenges, the challenges in Gaza are particularly acute and therefore worth additional scrutiny. Such challenges include prolonged periods of active conflict and Gaza being under siege since 2007, together with restrictions on mobility for both patients and healthcare professionals. Adding a seventh ‘Gaza specific’ building block became especially useful in allowing us to analyse data that was specific to Gaza, and was also especially helpful in devising recommendations for reconstructing the health system in Gaza.”

5. R2 asked for a more detailed discusison of the sample size, together with a reflection on whether the sampling approach yielded a representative sample.

We have now addressed this point by adding a new paragraph under the ‘study design’ heading of the methods section:

“The final list of participants included more frontline HCWs (doctors, nurses, paramedics, physiotherapists), who are naturally in larger proportion compared to HCWs in managerial and leadership positions and in NGOs, meaning the sample was somewhat representative of HCWs involved in MCI response. Also, the list included multiple HCWs from different surgical specialties which were the main specialties involved with the response to MCIs. In turn, and given that surgical specialties are often male-dominated, the list of participants we invited and those who participated were predominantly male.”

We have also added some sentences addressing this issue in the limitations section:

“Second, the study sample is missing HCWs from non-surgical specialties, e.g. internal medicine, pediatrics, and primary healthcare, whose insights and inputs are valuable when it comes to reconstructing the health in Gaza. This was due to the topic of interest of the study being response to MCIs, which mainly relates to HCWs in surgical specialties.”

6. R2 asks us to provide an indication of the extent of gender imbalance among HCWs in Gaza in relation to the gender imbalance of our sample.

This is an important point but because of a dearth of data on MOH gender breakdown, we are not able to provide a straightforward statistic. For example, the annual health report 2022 from the MOH-Gaza, which is the most recent annual report, does not show breakdown of HCWs by gender. In light of the absence of solid data, we have addressed this point by adding a new paragraph to the ‘study design’ heading of the methods section:

“The final list of participants included more frontline HCWs (doctors, nurses, paramedics, physiotherapists), who are naturally in larger proportion compared to HCWs in managerial and leadership positions and in NGOs, meaning the sample was somewhat representative of HCWs involved in MCI response. Also, the list included multiple HCWs from different surgical specialties which were the main specialties involved with the response to MCIs. In turn, and given that surgical specialties are often male-dominated, the list of participants we invited and those who participated were predominantly male.”

We have also added a sentence in the limitations section to address this:

“Also, it could be a reflection of the gender imbalance in the distribution of HCWs working in MCIs planning and response in the health system in Gaza, particularly that the response is dominated by surgical specialties which are often male-dominated.”

Minor Comments:

1. Define “building blocks” the first time it appears in the Data Analysis subsection.

This had been added and the six building blocks named:

“We therefore used the health system building blocks (30) – service delivery, human resources for health, leadership and governance, medical supplies, equipment and essential medicines, health information systems, and health financing – as guiding themes in the deductive content analysis through which we analysed the interviews.”

2. The figure’s readability could be improved. The authors may consider using white background blocks with black text to enhance contrast and make the information easier to read.

The figure has been improved and black and white used to enhance contrast and readability. We would like to note that the figure’s resolution decreases when uploaded to the submission portal so if it can be sent directly to the journal it would be better.

Once again, we would like to reiterate our gratitude to the reviewers and to the Editor for their helpful comments. We believe that addressing these comments has significantly strengthened the paper and hope that we have amply and adequately addressed all comments.

We look forward to hearing from you and of course remain ready to make any further changes and additions.

---

## [Editor Report · Decision Letter 1]

28 Jan 2026

Rebuilding Gaza's health system: a qualitative study of healthcare workers' experiences and lessons learned from responding to mass casualty incidents (2018–2021)

PONE-D-25-53940R1

Dear authors

We’re pleased to inform you that your manuscript has been judged scientifically suitable for publication and will be formally accepted for publication once it meets all outstanding technical requirements.

Kind regards,

Nikolaos Georgantzis.

Academic Editor

PLOS One

---

## [Editor Report · Acceptance letter]

PONE-D-25-53940R1

PLOS One

Dear Dr. Ismail,

I'm pleased to inform you that your manuscript has been deemed suitable for publication in PLOS One. Congratulations! Your manuscript is now being handed over to our production team.

Kind regards,

on behalf of

Prof. Nikolaos Georgantzis

Academic Editor

PLOS One